# Neurons Equipped with Intrinsic Plasticity Learn Stimulus Intensity Statistics

**Travis Monk**
Cluster of Excellence Hearing4all
University of Oldenburg
26129 Oldenburg, Germany
`travis.monk@uol.de`

**Cristina Savin**
IST Austria
3400 Klosterneuburg
Austria
`csavin@ist.ac.at`

**Jörg Lücke**
Cluster of Excellence Hearing4all
University of Oldenburg
26129 Oldenburg, Germany
`joerg.luecke@uol.de`

## Abstract

Experience constantly shapes neural circuits through a variety of plasticity mechanisms. While the functional roles of some plasticity mechanisms are well-understood, it remains unclear how changes in neural excitability contribute to learning. Here, we develop a normative interpretation of intrinsic plasticity (IP) as a key component of unsupervised learning. We introduce a novel generative mixture model that accounts for the class-specific statistics of stimulus intensities, and we derive a neural circuit that learns the input classes and their intensities. We will analytically show that inference and learning for our generative model can be achieved by a neural circuit with intensity-sensitive neurons equipped with a specific form of IP. Numerical experiments verify our analytical derivations and show robust behavior for artificial and natural stimuli. Our results link IP to non-trivial input statistics, in particular the statistics of stimulus intensities for classes to which a neuron is sensitive. More generally, our work paves the way toward new classification algorithms that are robust to intensity variations.

## 1 Introduction

Confronted with the continuous flow of experience, the brain takes amorphous sensory inputs and translates them into coherent objects and scenes. This process requires neural circuits to extract key regularities from their inputs and to use those regularities to interpret novel experiences. Such learning is enabled by a variety of plasticity mechanisms which allow neural networks to represent the statistics of the world. The most well-studied plasticity mechanism is synaptic plasticity, where the strength of connections between neurons changes as a function of their activity [1]. Other plasticity mechanisms exist and operate in tandem. One example is *intrinsic plasticity* (IP), where a neuron's response to inputs changes as a function of its own past activity. It is a challenge for computational neuroscience to understand how different plasticity rules jointly contribute to circuit computation.

While much is known about the contribution of Hebbian plasticity to different variants of unsupervised learning, including linear and non-linear sparse coding [2–5], ICA [6], PCA [7] or clustering [8–12], other aspects of unsupervised learning remain unclear. First, on the computational side, there are many situations in which the meaning of inputs should be invariant to its overall gain. For example, a visual scene's content does not depend on light intensity, and a word utterance should

be recognized irrespective of its volume. Current models do not explicitly take into account such gain variations, and often eliminate them using an *ad hoc* preprocessing step that normalizes inputs [8, 9, 13]. Second, on the biological side, the roles of other plasticity mechanisms such as IP, and their potential contributions to unsupervised learning, remain poorly understood.

IP changes the input-output function of a neuron depending on its past activity. Typically, IP is a homeostatic negative feedback loop that preserves a neuron's activation levels despite its changing input [14, 15]. There is no consensus on which quantities IP regulates, e.g. a neuron's firing rate, its internal Ca concentration, its spiking threshold, etc. In modeling work, IP is usually implemented as a simple threshold change that controls the mean firing rate, although some models propose more sophisticated rules that also constrain higher order statistics of the neuron's output [6, 16]. Functionally, while there have been suggestions that IP can play an important role in circuit function [6, 10, 11, 17], its role in unsupervised learning is still not fully understood.

Here we show that a neural network that combines specific forms of Hebbian plasticity and IP can learn the statistics of inputs with variable gain. We propose a novel generative model named *Product-Poisson-Gamma* (PPG) that explicitly accounts for class-specific variation in input gain. We then derive, from first principles, a neural circuit that implements inference and learning for this model. Our derivation yields a novel IP rule as a required component of unsupervised learning given gain variations. Our model is unique in that it directly links IP to the gain variations of the pattern to which a neuron is sensitive, which may be tested experimentally. Beyond neurobiology, the models provide a new class of efficient clustering algorithms that do not require data preprocessing. The learned representations also permit efficient classification from very little labeled data.

## 2 The Product-Poisson-Gamma model

Intensity can vary drastically across images although the features present in it are the same.[1] This variability constitutes a challenge for learning and is typically eliminated through a preprocessing stage in which the inputs are normalized [9]. While such preprocessing can make learning easier, *ad hoc* normalizations may be suboptimal, or may require additional parameters to be set by hand. More importantly, input normalization has the side-effect of losing information about intensity, which might have helped identify the features themselves. For instance, in computer vision objects of the same class are likely to have similar surface properties, resulting in a characteristic distribution of light intensities. Light intensities can therefore aid classification. In the neural context, the overall drive to neurons may vary, e.g. due to attentional gain modulation, despite the underlying encoded features being the same.

A principled way to address intensity variations is to explicitly model them in a generative model describing the data. Then we can use that generative model to derive optimal inference and learning for such data and map them to a corresponding neural circuit implementation. Let us assume the stimuli are drawn from one of $C$ classes, and let us denote a stimulus by $\vec{y}$. Given a stimulus / data point $\vec{y}$, we wish to infer the class $c$ that generated it (see Figure 1). Let $\vec{y}$ depend not only on the class $c$, but also on a continuous random variable $z$, representing the intensity of the stimulus, that itself depends on $c$ as well as some parameters $\theta$. Given these dependencies $\Pr(\vec{y}|c, z, \theta)$ and $\Pr(z|c, \theta)$, Bayes' rule specifies how to infer the class $c$ and hidden variable $z$ given an observation of $\vec{y}$:

$$\Pr(c, z|\vec{y}, \theta) = \frac{\Pr(\vec{y}|c, z, \theta)\Pr(z|c, \theta)\Pr(c|\theta)}{\sum_{c'} \int \Pr(\vec{y}|c', z', \theta)\Pr(z'|c', \theta)\Pr(c'|\theta)dz}. \tag{1}$$

We can obtain neurally-implementable expressions for the posterior if our data generative model is a mixture model with non-negative noise, e.g. a Poisson mixture model [9]. We extend the Poisson mixture model by including an additional statistical description of stimulus intensity. The Gamma distribution is a natural choice due to its conjugacy with the Poisson distribution. Let each of the $D$ elements in the vector $\vec{y}|z, c, \theta$ (e.g. pixels in an image) be independent and Poisson-distributed, let $z|c, \theta$ be Gamma-distributed, and let the prior of each class be uniform:

$$\Pr(\vec{y}|c, z, \theta) = \prod_{d=1}^{D} \mathrm{Pois}(y_d; zW_{cd}); \quad \Pr(z|c, \theta) = \mathrm{Gam}(z; \alpha_c, \beta_c); \quad \Pr(c|\theta) = \frac{1}{C}$$

where all $W$, $\alpha$, and $\beta$ represent the parameters of the model. To avoid ambiguity in scales, we constrain the weights of the model to sum to one, $\sum_d W_{cd} = 1$. We call this generative model a Product-Poisson-Gamma (PPG). While the multiplicative interaction between features and the intensity or gain variable is reminiscent of the Gaussian Scale Mixture (GSM) generative model, note that PPG has separate intensity distributions for each of the classes; each is a Gamma distribution with a (possibly unique) shape parameter $\alpha_c$ and rate parameter $\beta_c$. Furthermore, the non-gaussian observation noise is critical for deriving the circuit dynamics.

The model is general and flexible, yet it is sufficiently constrained to allow for closed-form joint posteriors. As shown in Appendix A, the joint posterior of the class and intensity is:

$$\Pr(c, z|\vec{y}, \theta) = \frac{\mathrm{NB}(\hat{y}; \alpha_c, \frac{1}{\beta_c+1}) \exp\left(\sum_d y_d \ln W_{cd}\right)}{\sum_{c'} \mathrm{NB}(\hat{y}; \alpha_{c'}, \frac{1}{\beta_{c'}+1}) \exp\left(\sum_d y_d \ln W_{c'd}\right)} \mathrm{Gam}(z; \alpha_c + \hat{y}, \beta_c + 1),$$

where $\hat{y} = \sum_d y_d$, and NB represents the negative binomial distribution.

We also obtain a closed-form expression of the posterior marginalized over $z$, which takes the form of a softmax function weighted by negative binomials:

$$\Pr(c|\vec{y}, \theta) = \frac{\mathrm{NB}(\hat{y}; \alpha_c, \frac{1}{\beta_c+1}) \exp\left(\sum_d y_d \ln W_{cd}\right)}{\sum_{c'} \mathrm{NB}(\hat{y}; \alpha_{c'}, \frac{1}{\beta_{c'}+1}) \exp\left(\sum_{d'} y_{d'} \ln W_{c'd'}\right)} \tag{2}$$

This is a straightforward generalization of the standard softmax, used for optimal learning in winner-take-all (WTA) networks [2,8,9,11] and WTA-based microcircuits [18]. Note that Eqn. 2 represents the optimal way to integrate evidence for class membership originating from stimulus intensity (parameterized by $\vec{\alpha}$ and $\vec{\beta}$) and pattern 'shape' (parameterized by $W$). If one of the two is not instructive, then the corresponding terms cancel out: if the patterns have identical shape ($W$ with identical rows), then the softmax drops out and only negative binomial terms remain, and if all pattern classes have the same intensity distribution, then the posterior reduces to the standard softmax function as in previous work [2, 8–11].

To facilitate the link to neural dynamics, Eqn. 2 can be simplified by approximating the negative binomial distribution as Poisson. In the limit that $\alpha_c \to \infty$ and the mean $\lambda_c \equiv \alpha_c/\beta_c = $ constant, the negative binomial distribution is:

$$\lim_{\alpha_c \to \infty, \alpha_c/\beta_c=\mathrm{const.}} \mathrm{NB}(\hat{y}; \alpha_c, \frac{1}{\beta_c+1}) = \mathrm{Pois}(\hat{y}; \frac{\alpha_c}{\beta_c}) \equiv \mathrm{Pois}(\hat{y}; \lambda_c).$$

In this limit, Eqn. 2 becomes:

$$\Pr(c|\vec{y}, \theta) \approx \frac{\exp(\sum_{d'} y_{d'} \ln(W_{cd'}\lambda_c) - \lambda_c)}{\sum_{c'} \exp(\sum_{d'} y_{d'} \ln(W_{c'd'}\lambda_{c'}) - \lambda_{c'})} \tag{3}$$

which can be evaluated by a neural network using soft-WTA dynamics [9].

## 3 Expectation-Maximization of PPG-generated data

As a starting point for deriving a biologically-plausible neural network for learning PPG-generated data, let us first consider optimal learning derived from the Expectation-Maximization (EM) algorithm [19]. Given a set of $N$ data points $\vec{y}^{(n)}$, we seek the parameters $\theta = \{\mathbf{W}, \boldsymbol{\lambda}\}$ that maximize the data likelihood given the PPG-model defined above. We use the EM formulation introduced in [20] and optimize the free-energy given by:

$$\mathcal{F}(\theta_t, \theta_{t-1}) = \sum_n \sum_{c'} \Pr(c'|\vec{y}^{(n)}, \theta_{t-1})(\ln \Pr(\vec{y}^{(n)}|c', \theta_t) + \ln \Pr(c'|\theta_t)) + H(\theta_{t-1}).$$

Here, $H(\theta_{t-1})$ is the Shannon entropy of the posterior as a function of the previous parameter values.

We can find the M-step update rules for the parameters of the model $\lambda_c$ and $W_{cd}$ by taking the partial derivative of $\mathcal{F}(\theta_t, \theta_{t-1})$ w.r.t. the desired parameter and setting it to zero. As shown in Appendix B, the resultant update rule for $\lambda_{c,t}$ is:

$$\frac{\partial \mathcal{F}(\theta_t, \theta_{t-1})}{\partial \lambda_{c,t}} = 0 \Rightarrow \lambda_{c,t} = \frac{\sum_n \Pr(c|\vec{y}^{(n)}, \theta_{t-1})\hat{y}^{(n)}}{\sum_n \Pr(c|\vec{y}^{(n)}, \theta_{t-1})} \tag{4}$$

The M-step update rules for the weights $W_{cd}$ are found by setting the corresponding partial derivative of $\mathcal{F}(\theta_t, \theta_{t\text{-}1})$ to zero, under the constraint that $\sum_d W_{cd} = 1$. Using Lagrange multipliers $\Lambda_c$ yields the following update rule (see Appendix B):

$$\frac{\partial \mathcal{F}(\theta_t, \theta_{t\text{-}1})}{\partial W_{cd,\text{t}}} + \frac{\partial}{\partial W_{cd,\text{t}}} \sum_{c'} \Lambda_{c'} \left( \sum_{d'} W_{c'd',\text{t}} - 1 \right) = 0$$

$$\Rightarrow W_{cd,\text{t}} = \frac{\sum_n y_d \Pr(c|\vec{y}^{(n)}, \theta_{t\text{-}1})}{\sum_d \sum_n y_d \Pr(c|\vec{y}^{(n)}, \theta_{t\text{-}1})}. \tag{5}$$

As numerical verification, Figure 1 illustrates the evolution of parameters $\lambda_c$ and $W_{cd}$ yielded by the EM algorithm on artificial data. Our artificial data set consists of four classes of rectangles on a grid of 10x10 pixels. Rectangles from different classes have different sizes and positions and are represented by a generative vector $W_c^{\text{gen}}$.

We generate a data set by drawing a large number $N$ of observations of $W_c^{\text{gen}}$, with each class equiprobable. We then draw a random variable $z$ from a Gamma distribution with parameters $\alpha_c$ and $\beta_c$ that depend on the class of each observation. Then, given $W_c^{\text{gen}}$ and $z$, we create a data vector $\vec{y}^{(n)}$ by adding Poisson noise to each pixel. With a set of $N$ data vectors $\vec{y}^{(n)}$, we then perform EM to find the parameters $W_{cd}$ and $\lambda_c$ that maximize the likelihood of the data set (at least locally). The E-step evaluates Equation 2 for each data vector, and the M-step evaluates Equations 4 and 5. Figure 1 shows that, after about five iterations, the EM algorithm returns the values of $W_{cd}$ and $\lambda_c$ that were used to generate the data set, i.e. the parameter values that maximize the data likelihood.

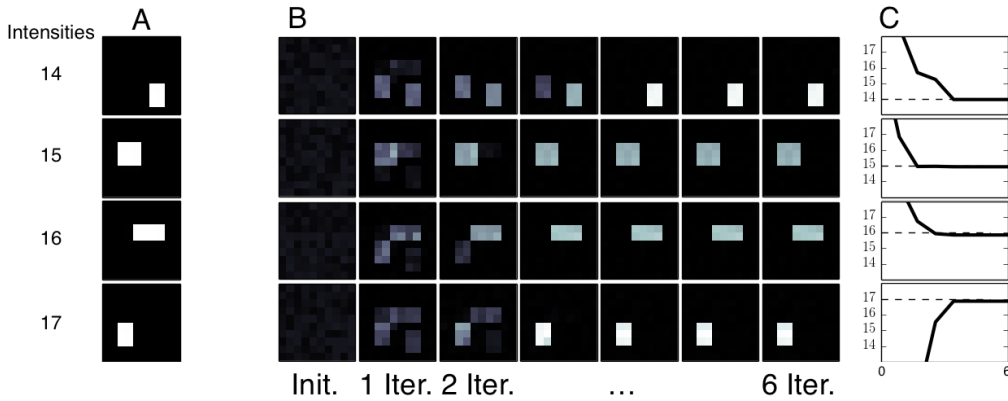

Figure 1: The evolution of model parameters yielded by the EM algorithm on artificial data. A: Four classes of rectangles represented by the vector $W_c^{\text{gen}}$, with the values of $\lambda_c$ for each class displayed to the left. B: Evolution of the parameters $W_{cd}$ for successive iterations of the EM algorithm. C: Evolution of the parameters $\lambda_c$, with dashed lines indicating the values from the generative model. The EM algorithm returns the values of $W_{cd}$ and $\lambda_c$ that were used to generate the data set, i.e. the parameter values that maximize the data likelihood. For these plots, we generated a data set of 2000 inputs. $W_c^{\text{gen}} = 100$ for white pixels and 1 for black pixels. The shape and rate parameters of the Gamma distributions, from the top class to the bottom, are $\alpha = [98, 112, 128, 144]$ and $\beta = [7, 7.5, 8, 8.5]$, giving $\lambda_c = \alpha_c/\beta_c = [14, 15, 16, 17]$.

## 4   Optimal neural learning for varying stimulus intensities

For PPG-generated data, the posterior distribution of the class given an observation is approximately the softmax function (or soft-WTA, Eqn. 3). Neural networks that implement the softmax function, usually via some form of lateral inhibition, have been extensively investigated [2, 8–11, 21]. Thus, inference in our model reduces to well-understood neural circuit dynamics.

The key remaining challenge is to analytically relate optimal learning as derived by EM to circuit plasticity. To map abstract random variables to neural counterparts, we consider a complete bipartite neural network, with the input layer corresponding to the observables $\mathbf{y}$ and the hidden layer representing the latent causes of the observables, i.e. classes.[2] The network is feedforward; each

neuron in the input layer connects to each neuron in the hidden layer via synaptic weights $W_{cd}$, where $c \in [1, C]$ indexes the $C$ hidden neurons and $d \in [1, D]$ indexes the $D$ input neurons.

Let each of the hidden neurons have a standard activity variable, $s_c$, and additionally an intrinsic parameter $\lambda_c$ that represents its excitability. Let the activity of each hidden neuron be given by Eqn. 2. The activity of each hidden neuron is then the posterior distribution for one particular class, given the inputs it receives from the input layer, its synaptic weights, and its excitability:

$$s_c = \frac{\exp(I_c)}{\sum_{c'} \exp(I_{c'})}; \qquad I_c = \sum_{d'} y_{d'} \ln(W_{cd'} \lambda_c) - \lambda_c.$$

The weights of the neural network $W_{cd}$ are plastic and change according to a Hebbian learning rule with synaptic scaling [22]:

$$\Delta W_{cd} = \epsilon_W (s_c y_d - s_c \lambda_c \bar{W}_c W_{cd}), \tag{6}$$

where $\epsilon_W$ is a small and positive learning rate, and $\bar{W}_c = \sum_d W_{cd}$.

The intrinsic parameters $\lambda_c$ are also plastic and change according to a similar learning rule:

$$\Delta \lambda_c = \epsilon_\lambda s_c (\sum_d y_d - \lambda_c), \tag{7}$$

where $\epsilon_\lambda$ is another small positive learning rate. This type of regulation of excitability is homeostatic in form, but differs from standard implementations in that the excitability changes not only depending on the neuron output, $s$, but also on the net input to the neuron (see also [17] for a formal link between $\sum_d y_d$ and average incoming inputs).

Appendix C shows that these online update rules enforce the desired weight normalization, with $\bar{W}_c$ converging to one. Assuming weight convergence, and assuming a small learning rate and a large set of data points, the weights and intrinsic parameters converge to (see [9] and Appendix C):

$$W_{cd}^{\mathrm{conv}} \approx \frac{\sum_n y_d^{(n)} s_c}{\sum_{d'} \sum_n y_d^{(n)} s_c}; \qquad \lambda_c^{\mathrm{conv}} = \frac{\sum_n s_c \hat{y}^{(n)}}{\sum_n s_c}.$$

Comparing these convergence expressions with the EM updates (Eqns. 5 and 4) and inserting the definition $s_c = \Pr(c|\vec{y}, \theta)$, we see that the neural dynamics given in Eqns. 6 and 7 have the same fixed points as optimal EM learning. The network can therefore find the parameter values that optimize the data likelihood using compact and neurally-plausible learning rules. Eqn. 6 is a standard form of Hebbian plasticity with synaptic scaling, while Eqn. 7 states how the excitability of hidden neurons should be governed by the gain of the inputs and the current to the neuron.

## 5   Numerical Experiments

To verify our analytical results, we first investigated learning in the derived neural network using data generated according to the PPG model. Figure 2 illustrates the evolution of parameters $\lambda_c$ and $W_{cd}$ yielded by the neural network on artificial data (the same as used for Figure 1). The neural network learns the synaptic weights and intrinsic parameters that were used to generate the data set, i.e. the parameter values that maximize the data likelihood.

Since our artificial data was PPG-generated, one can expect the neural network to learn the classes and intensities quickly and accurately. To test the neural network on more realistic data, we followed a number of related studies [8–12] and used the MNIST as a standard dataset containing different stimulus classes. The input to the network was 28x28 pixel images (converted to vectors) from the MNIST dataset. We present our results for the digits 0-3 for visual ease and simulation speed; our results on the full dataset are qualitatively similar. We added an offset of 1 to all pixels and rescaled them so that no pixel was greater than 1. The $\lambda_c$ were initialized to be the mean intensity of all digit classes as calculated from our modified MNIST training set. Each $W_{cd}$ was initialized as $W_{cd} \sim \mathrm{Pois}(W_{cd}; \mu_d) + 1$, where $\mu_d$ is the mean of each pixel over all classes and is calculated from our modified MNIST training set.

Figure 3 shows an example run using $C = 16$ hidden neurons. It shows the change in both neural weights and intrisic excitabilities $\lambda_c$ during learning. We observe that the weights change to represent the digit classes and converge relatively quickly (panels A, B). We verified that they sum to 1

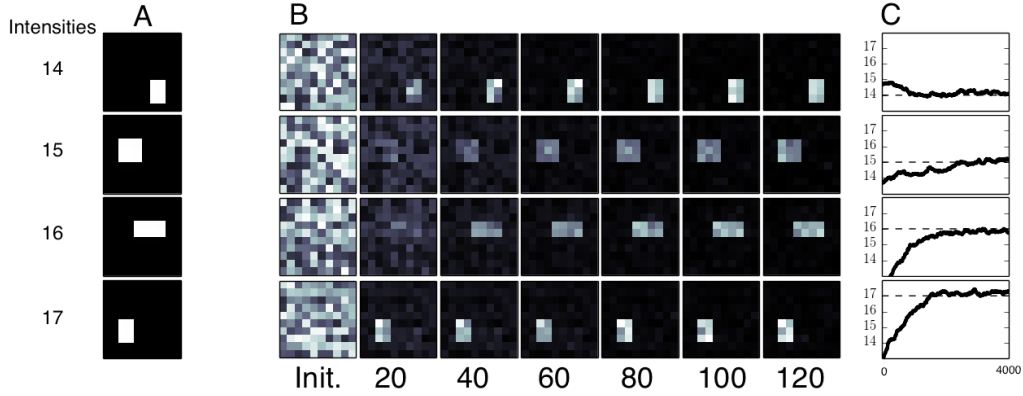

Figure 2: The evolution of model parameters yielded by the neural network on artificial data generated from the same model as that used in Figure 1. A: Four classes of rectangles with the values of $\lambda_c$ for each class displayed to the left. B: Evolution of the synaptic weights $W_{cd}$ that feed each hidden unit after 0, 20, 40, ..., 120 time steps, respectively. C: Evolution of the intrinsic parameters $\lambda_c$ over 4000 time steps, with dashed lines indicating the values from the generative model. The neural network returns the values of $W_{cd}$ and $\lambda_c$ that were used to generate the data set, i.e. the parameter values that maximize the data likelihood. For these plots, $\epsilon_W = \epsilon_\lambda = .005$, $D = 100$ (for a 10x10 pixel grid), $C = 4$, initialized weights were uniformly-distributed between .01 and .06, and initialized intrinsic parameters were uniformly-distributed between 10 and 20.

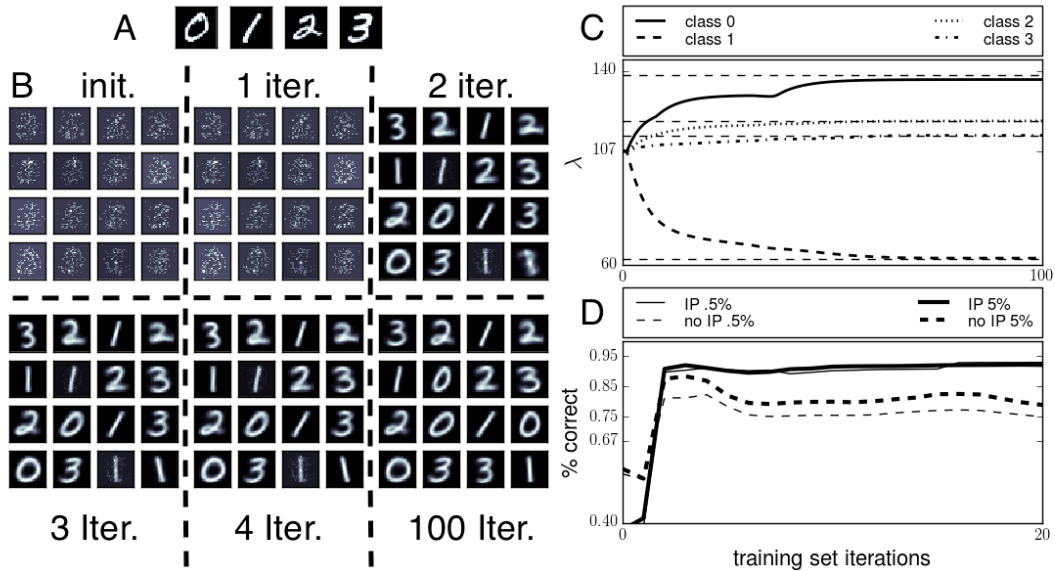

Figure 3: The neural network's performance on a reduced MNIST dataset (the digits 0 to 3). A: Representatives of the input digits. B: The network's synaptic weights during training. Each square represents the weights feeding one hidden neuron. Each box of 16 squares represents the weights feeding each of the $C = 16$ hidden neurons after initialization, and after subsequent iterations over the training set. The network learns different writing styles for different digits. C: The network learns the average intensities, i.e. the sum of the pixels in an image, of each class of digit in MNIST. Algorithms that impose *ad hoc* intensity normalization in their preprocessing cannot learn these intensities. The horizontal dashed lines are the average intensities of each digit, with 1 having the lowest overall luminance and 0 the largest. The average $\lambda_c$ for all hidden units representing a given digit converge to those ground truth values. D: The network's learned intensity differences improve classification performance. The percentage of correct digit classifications by a network with IP (solid lines) is higher than that by a network without IP (dashed lines). This result is robust to the number of iterations over the dataset and the number of labels used to calculate the Bayesian classifier used in [9].

for each class at convergence (not shown). We also observe that the network's IP dynamics allow it to learn the average intensities of each class of digit (panel C). The thin horizontal dashed lines are the true values for $\lambda_c$ as calculated from the MNIST test set using its ground-truth label information. IP modifies the network's excitability parameters $\lambda$ to converge to their true values. Our network is not only robust to variations in intensity, but learns their class-specific values.

A network that learns the excitability parameters $\lambda$ exhibits a higher classification rate than a network without IP (panel D). We computed the performance of the network derived in Sec. 4 on unnormalized data in comparison with a network without IP (all else being equal). As a performance measure we used the classification error (computed using the same Bayesian classifier as used in [9]). Classification success rates were calculated with very few labels, using $0.5\%$ (thin lines) and $5\%$ (thick lines) of labels in the training set (both settings for both networks). The classification performance of the network with IP outperforms that of the network without it. This result suggests that the differences in intensities in MNIST, albeit visually small, are sufficient to aid classification.

Finally, Figure 4 shows that the neural network can learn classes that differ only in their intensities. The dataset used for Figure 4 comprises 40000 images of two types of sphere: dull and shiny. The spheres were identical in shape and position, and we generated data points (i.e. images) under a variety of lighting conditions. On average, the shiny spheres were brighter ($\lambda_{\text{shiny}} \approx 720$) than the dull spheres ($\lambda_{\text{dull}} \approx 620$). The network represents the two classes in its learned weights and intensities. Algorithms that utilize *ad hoc* normalization preprocessing schemes would have serious difficulties learning input statistics for datasets of this kind.

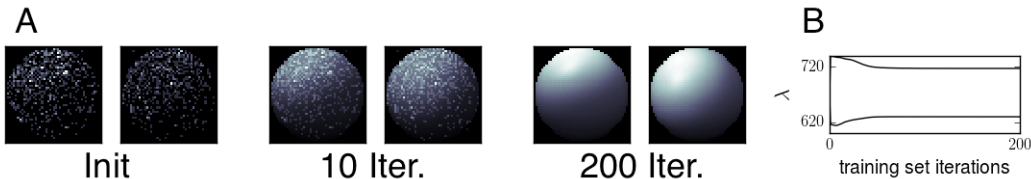

Figure 4: The neural network can learn classes that differ only in their intensities. The dataset consisted of either dull or shiny spheres. The network had $C = 2$ hidden neurons. A: Three pairs of squares represent the weights feeding each hidden neuron after initialization (leftmost pair), 10 iterations (center pair), and 200 iterations (rightmost pair) over the training set. Note the rightmost pair, particularly how the right sphere appears brighter than the left sphere. The right sphere corresponds to the shiny class and the left sphere to the dull class. B: Learned mean intensities as a function of iterations over the training set. The dull spheres have an average intensity of 620, and the shiny spheres 720. The network learns the classes and their average intensities, even when data points from different classes have the same sizes and positions.

## 6 Discussion

Neural circuit models are powerful tools for understanding neural learning and information processing. They have attracted attention as inherently parallel information processing devices for analog VLSI, a fast and power-efficient alternative to standard processor architectures [12, 23]. Much work has investigated learning with winner-take-all (WTA) type networks [2, 8–12, 18, 21, 24]. A subset of these studies [2, 8–11, 21] link synaptic plasticity in WTA networks to optimal learning, mostly using mixture distributions to model input stimuli [8–11, 21]. Our contribution expands on these results both computationally, by allowing for a robust treatment of variability in input gain, and biologically, by providing a normative justification for intrinsic plasticity during learning. Our analytical results show that the PPG-generative model is tractable and neurally-implementable, while our numerical results show that it is flexible and robust.

Our model provides a principled treatment of intensity variations, something ubiquitous in realistic datasets. As a result, it allows for robust learning without requiring normalized input data. This addresses the criticisms (see [10]) of earlier WTA-like circuits [8,9] that required normalized data. We found that explicitly accounting for intensity improves classification performance even for datasets that have been size-normalized (e.g. MNIST), presumably by providing an additional dimension for discriminating across latent features. Furthermore, we found that the learned representation of the MNIST data allows for good classification in a semi-supervised setting, when only a small fraction

of the data is labeled. Thus, our model provides a starting point for constructing novel clustering and classification algorithms following the general approach in [9].

The treatment of intensity as an explicit variable is not new. The well-investigated class of Gaussian Scale Mixtures (GSM) is built on that idea. Nonetheless, while GSM and PPG share some conceptual similarities, they are mathematically distinct. While GSMs assume 1) Gaussian distributed random variables and 2) a common scale variable [25], PPG assumes 1') Poisson observation noise and 2') class-specific scale variables. Consequently, none of the GSM results carry over to our work, and our PPG assumptions are critical for our derived intrinsic plasticity and Hebbian plasticity rules. It would be interesting to investigate a circuit analog of intensity parameter learning in a GSM. Since this class of models is known to capture many features of afferent sensory neurons, we might make more specific predictions concerning IP in V1. It would also be interesting to compare the classification performance of a GSM with that of PPG on the same dataset. The nature of the GSM generative model (linear combination of features with multiplicative gain modulation) makes it an unusual choice for a classification task. However, in principle, one could use a GSM to learn a representation of a dataset and train a classifier on it.

The optimal circuit implementation of learning in our generative model requires a particular form of IP. The formulation of IP is a phenomenological one, reflecting the biological observation that the excitability of a neuron changes in a negative feedback loop as a function of past activity [14, 15]. Mathematically, our model shares similarities with past IP models [6, 10, 17] with the important difference that the controlled variable is the *input current*, rather than the output firing rate. Since the two quantities are closely related, we expect it will be difficult to directly disambiguate between IP models experimentally. Nonetheless, our model makes potentially testable predictions in terms of the functional role of IP, by directly linking the excitability of individual neurons to nontrivial statistics of their inputs, namely their average intensity under a Gamma distribution. Since past IP work invariably assumes the target excitability is a fixed parameter, usually shared across neurons, the link between neural excitability and real world statistics is very specific to our model and potentially testable experimentally. Furthermore, our work provides a computational rationale for the dramatic variations in excitability across neurons, even within a local cortical circuit, which could not be explained by traditional models.

The functional role for IP identified here complements previous proposals linking the regulation of neuronal excitability to learning priors [11] or as posterior constraints [10, 26]. Ultimately, it is likely that the role of IP is manifold. Recent theoretical work suggests that the net effect of inputs on neural excitability may arise as a complex interaction between several forms of IP, some homeostatic and others not [17]. Furthermore, different experimental paradigms may preferentially expose one IP process over the others, which would explain the confusion within the literature on the exact nature of biological IP. Taken together, these models point to a fundamental role of IP for circuit computation in a variety of setups. Given its many possible roles, any approach based on first principles is valuable, as it tightly connects IP to concrete stimulus properties in a way that can translate into better-constrained experiments.

**Acknowledgements.** We acknowledge funding by the DFG within the Cluster of Excellence EXC 1077/1 (Hearing4all) and by grant LU 1196/5-1 (JL and TM) and the People Programme (Marie Curie Actions) of the European Union's Seventh Framework Programme (FP7/2007-2013) under REA grant agreement no. 291734 (CS).

## Footnotes

[1]We use images as inputs and intensity as a measure of input gain as a running example. Our arguments apply regardless of the type of sensory input, e.g. the volume of sound or the concentration of odor.

[2]The number of hidden neurons does not necessarily need to equal the number of classes; see Figure 3.

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
