[Supplementary Material]

## Appendix A: Derivation of Posterior for PPG Data

Restating Bayes' rule (Equation 1):

$$\Pr(c, z|\vec{y}, \theta) = \frac{\Pr(\vec{y}|c, z, \theta)\Pr(z|c, \theta)\Pr(c|\theta)}{\sum_{c'} \int \Pr(\vec{y}|c', z', \theta)\Pr(z'|c', \theta)\Pr(c'|\theta)dz}.$$

Defining $\Pr(c|\theta) = 1/C$, $\Pr(c|\theta)$ drops:

$$\Pr(c, z|\vec{y}, \theta) = \frac{\Pr(\vec{y}|c, z, \theta)\Pr(z|c, \theta)}{\sum_{c'} \int \Pr(\vec{y}|c', z', \theta)\Pr(z'|c', \theta)dz}.$$

Inserting the likelihood and prior of $z$:

$$\Pr(\vec{y}|c, z, \theta) = \prod_{d=1}^{D} \text{Pois}(y_d; zW_{cd}); \quad \Pr(z|c, \theta) = \text{Gam}(z; \alpha_c, \beta_c),$$

Bayes' rule becomes:

$$\Pr(\vec{y}|c, z, \theta) = \frac{\prod_d \frac{(zW_{cd})^{y_d}\exp(-zW_{cd})}{y_d!}\frac{z^{\alpha_c-1}\exp(-z\beta_c)\beta_c^{\alpha_c}}{\Gamma(\alpha_c)}}{\sum_{c'} \int \prod_d \frac{(z_{c'}W_{c'd})^{y_d}\exp(-z_{c'}W_{c'd})}{y_d!}\frac{z_{c'}^{\alpha_{c'}-1}\exp(-z_{c'}\beta_{c'})\beta_{c'}^{\alpha_{c'}}}{\Gamma(\alpha_{c'})}dz_{c'}}$$

$$= \frac{(\prod_d W_{cd}^{y_d})z^{\sum_d y_d}\exp(-z\sum_d W_{cd})z^{\alpha_c-1}\exp(-z\beta_c)\beta_c^{\alpha_c}\Gamma(\alpha_c)^{-1}}{\sum_{c'}(\prod_d W_{cd}^{y_d})\int z^{\sum_d y_d}\exp(-z\sum_d W_{cd})z^{\alpha_c-1}\exp(-z\beta_c)\beta_c^{\alpha_c}\Gamma(\alpha_c)^{-1}dz_{c'}}.$$

Imposing the constraint $\sum_d W_{cd} = 1$ and letting $\hat{y} = \sum_d y_d$:

$$= \frac{(\prod_d W_{cd}^{y_d})z^{\hat{y}}\exp(-z)z^{\alpha_c-1}\exp(-z\beta_c)\beta_c^{\alpha_c}\Gamma(\alpha_c)^{-1}}{\sum_{c'}(\prod_d W_{cd}^{y_d})\int z^{\hat{y}}\exp(-z)z^{\alpha_c-1}\exp(-z\beta_c)\beta_c^{\alpha_c}\Gamma(\alpha_c)^{-1}dz_{c'}}$$

$$= \frac{(\prod_d W_{cd}^{y_d})z^{\hat{y}+\alpha_c-1}\exp(-z(\beta_c+1))\beta_c^{\alpha_c}\Gamma(\alpha_c)^{-1}}{\sum_{c'}(\prod_d W_{c'd}^{y_d})\int z_{c'}^{\hat{y}+\alpha_{c'}-1}\exp(-z_{c'}(\beta_{c'}+1))\beta_{c'}^{\alpha_{c'}}\Gamma(\alpha_{c'})^{-1}dz_{c'}}.$$

We can get rid of the integral by introducing the factors $(\beta_c+1)^{\hat{y}+\alpha_c}$ and $\Gamma(\hat{y}+\alpha_c)^{-1}$:

$$= \frac{(\prod_d W_{cd}^{y_d})\frac{\beta_c^{\alpha_c}}{(\beta_c+1)^{\hat{y}+\alpha_c}}\frac{\Gamma(\hat{y}+\alpha_c)}{\Gamma(\alpha_c)}z^{\hat{y}+\alpha_c-1}\exp(-z(\beta_c+1))\frac{(\beta_c+1)^{\hat{y}+\alpha_c}}{\Gamma(\hat{y}+\alpha_c)}}{\sum_{c'}(\prod_d W_{c'd}^{y_d})\frac{\beta_c^{\alpha_c}}{(\beta_c+1)^{\hat{y}+\alpha_c}}\frac{\Gamma(\hat{y}+\alpha_c)}{\Gamma(\alpha_c)}\int z^{\hat{y}+\alpha_c-1}\exp(-z(\beta_c+1))\frac{(\beta_c+1)^{\hat{y}+\alpha_c}}{\Gamma(\hat{y}+\alpha_c)}dz},$$

and recognizing the integrand as a Gamma distribution, which must integrate to 1. The corresponding term in the numerator is also a Gamma distribution:

$$= \frac{(\prod_d W_{cd}^{y_d})\frac{\beta_c^{\alpha_c}}{(\beta_c+1)^{\hat{y}+\alpha_c}}\frac{\Gamma(\hat{y}+\alpha_c)}{\Gamma(\alpha_c)}}{\sum_{c'}(\prod_d W_{c'd}^{y_d})\frac{\beta_{c'}^{\alpha_{c'}}}{(\beta_{c'}+1)^{\hat{y}+\alpha_{c'}}}\frac{\Gamma(\hat{y}+\alpha_{c'})}{\Gamma(\alpha_{c'})}}\text{Gam}(z; \alpha_c+\hat{y}, \beta_c+1)$$

Multiplying the numerator and denominator by $(\hat{y}!)^{-1}$:

$$= \frac{(\prod_d W_{cd}^{y_d})\frac{\beta_c^{\alpha_c}}{(\beta_c+1)^{\hat{y}+\alpha_c}}\frac{\Gamma(\hat{y}+\alpha_c)}{\Gamma(\alpha_c)\hat{y}!}}{\sum_{c'}(\prod_d W_{c'd}^{y_d})\frac{\beta_{c'}^{\alpha_{c'}}}{(\beta_{c'}+1)^{\hat{y}+\alpha_{c'}}}\frac{\Gamma(\hat{y}+\alpha_{c'})}{\Gamma(\alpha_{c'})\hat{y}!}}\text{Gam}(z; \alpha_c+\hat{y}, \beta_c+1),$$

we can now recognize the ratios in the numerator and denominator as negative binomial distributions. Thus Equation 1 can be written as:

$$\Pr(c, z|\vec{y}, \theta) = \frac{(\prod_d W_{cd}^{y_d})\text{NB}(\hat{y}; \alpha_c, \frac{1}{\beta_c+1})}{\sum_{c'}(\prod_d W_{c'd}^{y_d})\text{NB}(\hat{y}; \alpha_{c'}, \frac{1}{\beta_{c'}+1})}\text{Gam}(z; \alpha_c+\hat{y}, \beta_c+1).$$

We can now easily obtain $\Pr(c|\vec{y}, \theta)$ by integrating $\Pr(c, z|\vec{y}, \theta)$ over $z$:

$$\Pr(c|\vec{y}, \theta) = \frac{(\prod_d W_{cd}^{y_d})\text{NB}(\hat{y}; \alpha_c, \frac{1}{\beta_c+1})}{\sum_{c'}(\prod_d W_{c'd}^{y_d})\text{NB}(\hat{y}; \alpha_{c'}, \frac{1}{\beta_{c'}+1})}$$

$$= \frac{\text{NB}(\hat{y}; \alpha_c, \frac{1}{\beta_c+1})\exp\left(\sum_d y_d \ln W_{cd}\right)}{\sum_{c'}\text{NB}(\hat{y}; \alpha_{c'}, \frac{1}{\beta_{c'}+1})\exp\left(\sum_d y_d \ln W_{c'd}\right)},$$

which is our claimed expression in Equation 2.

## Appendix B: Derivation of M-Step Update Rules

Expectation-Maximization (EM) maximizes a lower bound of the log-likelihood called the free energy $\mathcal{F}(\theta_t, \theta_{t-1})$, which is a function of the parameter values from the previous and current iteration of EM:

$$\mathcal{F}(\theta_t, \theta_{t-1}) = \sum_n \sum_{c'} \Pr(c'|\vec{y}^{(n)}, \theta_{t-1})(\ln \Pr(\vec{y}^{(n)}|c', \theta_t) + \ln \Pr(c'|\theta_t)) + H(\theta_{t-1}).$$

where $H(\theta_{t-1})$ is the Shannon entropy as a function of the old parameter values only.

The M-step update rule for the parameters $\lambda_c$ is found by taking the partial derivative of the free energy and setting it to zero:

$$\frac{\partial \mathcal{F}(\theta_t, \theta_{t-1})}{\partial \lambda_{c,t}} = 0. \tag{8}$$

The partial derivative of all terms in the sum on $c'$ are zero, except for $c' = c$. Also, the Shannon entropy is a function of the old parameter values only. Thus, Equation 8 becomes:

$$0 = \frac{\partial}{\partial \lambda_{c,t}} \sum_n \Pr(c|\vec{y}^{(n)}, \theta_{t-1})(\ln \Pr(\vec{y}^{(n)}|c, \theta_t) + \ln \Pr(c|\theta_t))$$

$$= \sum_n \Pr(c|\vec{y}^{(n)}, \theta_{t-1}) \left( \frac{\frac{\partial}{\partial \lambda_{c,t}} \Pr(\vec{y}^{(n)}|c, \theta_t)}{\Pr(\vec{y}^{(n)}|c, \theta_t)} + \frac{\frac{\partial}{\partial \lambda_{c,t}} \Pr(c|\theta_t)}{\Pr(c|\theta_t)} \right).$$

Since the prior $\Pr(c|\theta_t) = 1/C$ is independent of $\lambda_{c,t}$, its derivative is zero. The likelihood of a data point is:

$$\Pr(\vec{y}^{(n)}|c, \theta_t) = \int \Pr(\vec{y}^{(n)}|z, c, \theta_t) \Pr(z|c, \theta_t) dz$$

$$= \int \left( \prod_{d=1}^{D} \mathrm{Pois}(y_d; zW_{cd}) \right) \mathrm{Gam}(z; \alpha_{c,t}, \beta_{c,t-1}) dz.$$

As shown in Appendix A, this integral is tractable:

$$\Pr(\vec{y}^{(n)}|c, \theta_t) = \left( \prod_{d=1}^{D} \frac{(W_{cd,t})^{y_d^{(n)}}}{y_d^{(n)}!} \right) (\hat{y}^{(n)}!) \mathrm{NB}(\hat{y}^{(n)}; \alpha_{c,t}, \beta_{c,t-1}).$$

In the limit that $\alpha_{c,t} \to \infty$ while $\alpha_{c,t}/\beta_{c,t-1}$ is held constant, the likelihood of a data point simplifies to:

$$\Pr(\vec{y}^{(n)}|c, \theta_t) \approx \left( \prod_{d=1}^{D} \frac{(W_{cd,t})^{y_d^{(n)}}}{y_d^{(n)}!} \right) \lambda_{c,t}^{\hat{y}^{(n)}} \exp(-\lambda_{c,t}).$$

Its derivative with respect to $\lambda_{c,t}$ has a compact form:

$$\frac{\partial}{\partial \lambda_{c,t}} \Pr(\vec{y}^{(n)}|c, \theta_t) = \Pr(\vec{y}^{(n)}|c, \theta_t) \left( \frac{\hat{y}^{(n)}}{\lambda_{c,t}} - 1 \right).$$

Equation 8 can then be written as:

$$\sum_n \Pr(c|\vec{y}^{(n)}, \theta_{t-1}) \left( \frac{\hat{y}^{(n)}}{\lambda_{c,t}} - 1 \right) = 0.$$

Rearranging:

$$\lambda_{c,t} = \frac{\sum_n \Pr(c|\vec{y}^{(n)}, \theta_{t-1}) \hat{y}^{(n)}}{\sum_n \Pr(c|\vec{y}^{(n)}, \theta_{t-1})}.$$

The update rule for the weights $W_{cd}$ are also found analogously, except for the presence of the constraint that $\sum_d W_{cd} = 1$. This constraint is enforced by introducing Lagrangian multipliers $\Lambda_c$:

$$\frac{\partial \mathcal{F}(\theta_t, \theta_{t-1})}{\partial W_{cd,t}} + \frac{\partial}{\partial W_{cd,t}} \sum_{c'} \Lambda_{c'} \left( \sum_{d'} W_{c'd',\mathrm{new}} - 1 \right) = 0. \tag{9}$$

The partial derivative of all terms in both sums on $c'$ are zero, except for $c' = c$. Also, the Shannon entropy is a function of the old parameter values only. The derivative of the free energy is then:

$$\frac{\partial \mathcal{F}(\theta_{\text{t}}, \theta_{\text{t-1}})}{\partial W_{cd,\text{t}}} = \sum_n \Pr(c|\vec{y}^{(n)}, \theta_{\text{t-1}}) \left( \frac{\frac{\partial}{\partial W_{cd,\text{t}}} \Pr(\vec{y}^{(n)}|c, \theta_{\text{t}})}{\Pr(\vec{y}^{(n)}|c, \theta_{\text{t}})} + \frac{\frac{\partial}{\partial W_{cd,\text{t}}} \Pr(c|\theta_{\text{t}})}{\Pr(c|\theta_{\text{t}})} \right).$$

Since $\Pr(c|\theta_{\text{t}}) = 1/C$ is independent of the weights, its derivative is zero. The derivative of $\Pr(\vec{y}^{(n)}|c, \theta_{\text{t}})$ has a compact form:

$$\frac{\partial}{\partial W_{cd,\text{t}}} \Pr(\vec{y}^{(n)}|c, \theta_{\text{t}}) = \Pr(\vec{y}^{(n)}|c, \theta_{\text{t}}) \left( \frac{y_d}{W_{cd,\text{t}}} \right),$$

so the derivative of the free energy is:

$$\frac{\partial \mathcal{F}(\theta_{\text{t}}, \theta_{\text{t-1}})}{\partial W_{cd,\text{t}}} = \sum_n \Pr(c|\vec{y}^{(n)}, \theta_{\text{t-1}}) \left( \frac{y_d^{(n)}}{W_{cd,\text{t}}} \right).$$

The partial derivative of all terms in the sum on $d'$ are zero, except for $d' = d$. Equation 9 is then:

$$\sum_n \left( \frac{y_d}{W_{cd,\text{t}}} \right) \Pr(c|\vec{y}^{(n)}, \theta_{\text{t-1}}) + \Lambda_c = 0. \tag{10}$$

Multiplying through by $W_{cd,\text{t}}$, summing over $d$, and letting $\sum_d W_{cd,\text{t}} = 1$, we find $\Lambda_c$:

$$\Lambda_c = -\sum_d \sum_n \Pr(c|\vec{y}^{(n)}, \theta_{\text{t-1}}) y_d^{(n)}.$$

Inserting $\Lambda_c$ into Equation 10 and rearranging for $W_{cd,\text{t}}$:

$$W_{cd,\text{t}} = \frac{\sum_n y_d \Pr(c|\vec{y}^{(n)}, \theta_{\text{t-1}})}{\sum_{d'} \sum_n y_{d'} \Pr(c|\vec{y}^{(n)}, \theta_{\text{t-1}})}.$$

This is the same updating rule for the weights as that derived in Keck et. al [9]. Notice that if we sum $W_{cd,\text{t}}$ over $d$, the sum must be 1 as required.

## Appendix C: Neural Network Learning Approximates EM

If our neural network's synaptic weights are normalized at convergence, then Keck et. al. [9] showed that those weights approximate those given by the EM algorithm for PPG data. Here, we only show that the sum of the weights for each hidden unit $\bar{W}_c \equiv \sum_d W_{cd}$ converges to 1, and refer interested readers to the complete proof in [9].

Recall the Hebbian plasticity rule for the synapse connecting input neuron $d$ to hidden neuron $c$:

$$\Delta W_{cd} = \epsilon_W (s_c y_d - s_c \lambda_c \bar{W}_c W_{cd}).$$

Summing both sides over $d$:

$$\Delta \bar{W}_c = \epsilon_W (s_c \hat{y} - s_c \lambda_c \bar{W}_c^2).$$

Assume that the weights have converged, and let the network observe a batch of $N$ data points. The change in $\bar{W}_c$ given the batch of $N$ data points is:

$$\Delta \bar{W}_c^{(N)} = \frac{1}{N} \sum_n \epsilon_W (s_c^{(n)} \hat{y}^{(n)} - s_c^{(n)} \lambda_c \bar{W}_c^2).$$

Assuming that the inputs $\vec{y}^{(n)}$ are drawn from a stationary distribution $\Pr(\vec{y}^{(n)})$, and assuming a small learning rate and a large batch size, we can accurately approximate the sum with an expectation:

$$\Delta \bar{W}_c^{(N)} \approx \epsilon_W \left( \langle s_c \hat{y} \rangle_{\Pr(\vec{y})} - \lambda_c \bar{W}_c^2 \langle s_c \rangle_{\Pr(\vec{y})} \right). \tag{11}$$

Inserting $s_c = \Pr(c|\vec{y}, \theta)$, the left expectation may be written as:

$$\langle s_c \hat{y} \rangle_{\Pr(\vec{y})} = \sum_{\vec{y}} \hat{y} \Pr(c|\vec{y}, \theta) \Pr(\vec{y}) = \sum_{\vec{y}} \hat{y} \frac{\Pr(c, \vec{y}|\theta)}{\Pr(\vec{y}|\theta)} \Pr(\vec{y}).$$

If the true data distribution is the same as the distribution learned by the model, then $\Pr(\vec{y}|\theta)$ and $\Pr(\vec{y})$ cancel:

$$\langle s_c \hat{y} \rangle_{\Pr(\vec{y})} = \Pr(c|\theta) \sum_{\vec{y}} \hat{y} \Pr(\vec{y}|c, \theta). \tag{12}$$

We can rewrite the sum as a conditional expectation:

$$\sum_{\vec{y}} \hat{y} \Pr(\vec{y}|c, \theta) = \sum_{\hat{y}} \hat{y} \sum_{\sum_d \vec{y} = \hat{y}} \Pr(\vec{y}|c, \theta) = \sum_{\hat{y}} \hat{y} \Pr(\hat{y}|c, \theta) = \langle \hat{y} \rangle_{\Pr(\hat{y}|c, \theta)}.$$

Using the tower property of conditional expectations and evaluating them for our generative model:

$$\langle \hat{y} \rangle_{\Pr(\hat{y}|c, \theta)} = \left\langle \langle \hat{y} \rangle_{\Pr(\hat{y}|z, c, \theta)} \right\rangle_{\Pr(z|c, \theta)} = \left\langle z \bar{W}_c \right\rangle_{\Pr(z|c, \theta)} = \bar{W}_c \lambda_c.$$

Inserting $\bar{W}_c \lambda_c$ for the sum in Equation 12:

$$\langle s_c \hat{y} \rangle_{\Pr(\vec{y})} \approx \Pr(c|\theta) \bar{W}_c \lambda_c.$$

The right expectation in Equation 11 is:

$$\langle s_c \rangle_{\Pr(\vec{y})} = \sum_{\vec{y}} \Pr(c|\vec{y}, \theta) \Pr(\vec{y}) = \sum_{\vec{y}} \frac{\Pr(\vec{y}|c, \theta) \Pr(c|\theta)}{\Pr(\vec{y}|\theta)} \Pr(\vec{y})$$

If the true data distribution is the same as the distribution learned by the model, then $\Pr(\vec{y}|\theta)$ and $\Pr(\vec{y})$ cancel:

$$\langle s_c \rangle_{\Pr(\vec{y})} = \Pr(c|\theta) \sum_{\vec{y}} \Pr(\vec{y}|c, \theta) = \Pr(c|\theta).$$

Inserting our expressions for $\langle s_c \hat{y} \rangle_{\Pr(\vec{y})}$ and $\langle s_c \rangle_{\Pr(\vec{y})}$ into Equation 11:

$$\Delta \bar{W}_c^{(N)} \approx \epsilon_W \Pr(c|\theta) \lambda_c \bar{W}_c \left(1 - \bar{W}_c\right).$$

This expression has stationary points at $\bar{W}_c = 1$ and 0. The stationary point at 1 is stable, while the stationary point at 0 is unstable. If the weights are initialized to be positive and the learning rate is sufficiently small, $\bar{W}_c$ converges to 1.

**Intrinsic Parameters**

Recall the learning rule for the intrinsic parameter of hidden neuron $c$:

$$\Delta \lambda_c = \epsilon_\lambda (s_c \hat{y} - s_c \lambda_c).$$

Consider the change in $\lambda_c$ given a batch of $N$ data points. Again assuming that the inputs are drawn from a stationary distribution, and assuming a small learning rate and large batch size, we can approximate $\Delta \lambda_c^{(N)}$ with expectations:

$$\Delta \lambda_c^{(N)} \approx \epsilon_\lambda (\langle s_c \hat{y} \rangle_{\Pr(\hat{y})} - \lambda_c \langle s_c \rangle_{\Pr(\hat{y})}).$$

This equation has a stable stationary point at:

$$\lambda_c = \frac{\langle s_c \hat{y} \rangle_{\Pr(\hat{y})}}{\langle s_c \rangle_{\Pr(\hat{y})}}.$$

Comparing this with Equation 4:

$$\lambda_{c,t} = \frac{\sum_n \Pr(c|\vec{y}^{(n)}, \theta_{t-1}) \hat{y}^n}{\sum_n \Pr(c|\vec{y}^{(n)}, \theta_{t-1})},$$

we see that the intrinsic parameters achieve stability when they approximate the expression yielded by the EM algorithm.