[Reviews · NeurIPS 2016]

Reviewer 1

Summary

This paper addresses the question of how neurons could optimally detect an object, independently of the contrast on the visual scene. The paper starts by deriving a generative model, the Product-Poisson-Gamma model, which explicitly describes gain invariance with respect to visual stimuli (although in principle generalisable to other stimuli modalities, e.g., auditory or olfactory). Learning the objects (“classes”) independently of the contrasts was accomplished through Expectation-Maximization. As for the neural implementation, the authors showed that intrinsic plasticity is able to learn the classes in a contrast-invariant manner.

Qualitative Assessment

The paper presents a convincing and novel solution to a biologically relevant question, which is still under debate in neuroscience: contrast-invariance computation. The paper is clearly written and acknowledges past solutions to the same problem. Some minor criticisms mostly about the presentation of the work: -the paper states that is provides a neural circuit that implements contrast-invariance. Given the biological nature of the paper, it would be less confusing to the biologically-inclined researcher for the paper to emphasise that these neural circuits are feedforward; -all figures should be more clearly labeled (label axes in convergence plots, label iterations on Fig3 left, label classes on Fig4 left); -Fig1 seems to have a inversion between row 3 and row 4; -define abbreviation WTA when first introduced. It is only defined in discussion.

Confidence in this Review

2-Confident (read it all; understood it all reasonably well)


Reviewer 2

Summary

This is a very interesting piece of work. The authors develop theory for providing a role to intrinsic plasticity, that of learning g the intensity of stimulus, which they further validate in simulations.

Qualitative Assessment

There are a few suggestions for improvement: Major comment: Why only training with 4 digits of MNIST and not the whole database? Please comment on intensity differences of the data in MNIST. It would be important to clearly show that you can achieve learning irrespective to the intensity variance as per the motivation of the paper, while other techniques don't. This would make the potential of the work much higher. Right now, it seems that images can be learned based on their intensity and that restricts the interest on the paper. Minor comments: 1. It mention that the hypothesis on IP can be experimentally verified. Discuss how. 2. Figures need improvement, digits are too small, the time dimension is not clear and seems that in Fig 1 intensities 16 and 17 are inverted. Similar comments for Fig 2 ( with 17 and 14). The caption of Figure 3 can be also improved, again letters are too small.

Confidence in this Review

2-Confident (read it all; understood it all reasonably well)


Reviewer 3

Summary

Neurons with Intrinsic Plasticity Learn Stimulus Statistics In the paper, the author(s) build a generative product-poisson-gamma model to recognize images with wide dynamic range, by extending a previously used Poisson mixture model. An equivalent neural circuit model with unsupervised learning was also proposed to implement EM-like algorithms to find the model parameters in generative model. In the learning rule, the synaptic weights are changed according to a Hebb like rule with synaptic scaling, and intrinsic parameters lambda (Eq.7 in the paper) are updated depending on the net input. It is very interesting that the fixed points of the network updating parameters can have the same form with EM algorithm. However, I have the following questions about this paper: 1. In the paper, the author claims that the parameter lambda corresponds to a neuron's intrinsic parameter, and updating of lambda implies to plasticity of intrinsic parameters. For me, I think here lambda is like a input gain control parameter in the neuron model, as it controls the input current that leads to saturation of the neuron. This is a very abstract intrinsic parameter, is it suitable to name it as intrinsic plasticity? 2. The number of neurons configured in the hidden layer. For me, I think in the model, if the number of hidden neuron equals to the classes in input data, then the network may have a good classification ability. While this is not always the case in reality, what would happen to a network if the number of hidden neuron does not equal to the number of classes in input data? Will the classification performance would degrade? 3. Time scale. It can be seen that, the synaptic weights and intrinsic parameters are updated at the same time, that is, the updating time constant would be the same. However, contrast of natural images and input pattern would not likely change with a same time constant in reality. So, what would happen in your model if synaptic weights and intrinsic parameter are updated with different time scale?

Qualitative Assessment

This is a theoretical beautiful work. It builds a neural circuit model with unsupervised learning to get an theoretical optimal fixed point, which is highly corresponded to EM learning algorithm result. This work considers the effect of contrast variation by extending a previously Poisson mixture model. The main concern is, what would happen when the hidden neuron number does not equals to the pattern class number, e.g., the number of hidden neuron is larger than the pattern classes.

Confidence in this Review

2-Confident (read it all; understood it all reasonably well)


Reviewer 4

Summary

The authors introduce a novel generative model that incorporates explicit modelling of stimulus intensity, based on Poisson distributed observations. Optimal learning is then derived by maximisation of the free energy. These update rules for the parameters of the model, accounting for class and intensity, are then translated into unsupervised Hebbian learning rules that implement a form of Intrinsic plasticity in a neural network. The authors provide a convincing and precise mathematical construction, and they extensively show the power of their method: first, by using artificial data as a proof of concept, then by using the MNIST dataset of hand-written digits and finally by showing how the network successfully classifies objects that differ only in intensity, by using images of dull and shiny spheres. The work seems very promising and represents one of the first attempts to learn stimulus intensity.

Qualitative Assessment

The manuscript is very clear and well written. The introduction gives already a very good understanding: what is the problem, what is available in the literature and what is the proposed solution and why it is useful. The reader is then carefully walked through the details of proposed solution and of the neural network implementation. Finally, to corroborate the theoretical derivation, a thorough study on three different datasets is carried on, including both simulated and real data. My only observation is about Figure 1, where centre plots and right plots of the cases lambda=16,17 are swapped. In general, the readability of all the 4 figures could improved, for instance by increasing the font size or adding text.

Confidence in this Review

2-Confident (read it all; understood it all reasonably well)


Reviewer 5

Summary

This paper presents a novel model called PPG: product poisson gamma model that characterizes intrinsic plasticity of neurons by considering the contrast information. The model learns best in a semi supervised fashion. The authors present sufficiently variant experiments to test their derivations.

Qualitative Assessment

The model and accompanying experiments are nicely presented. The topic and connections to previous literature has been nicely motivated. It would have been a stronger paper if PPG was compared to GSM models on the same datasets -- then it would be clear to the readers in what aspects these models are similar/different. What about real world images -- like cifar100 or imagenet? How do you think this model will perform? This is an important and timely topic that could potentially better performance on current deep learning architectures.

Confidence in this Review

2-Confident (read it all; understood it all reasonably well)


Reviewer 6

Summary

This paper proposes a model and develops learning rules that can account for gain variations. Part of the learning rules can be interpreted as implementing one form of intrinsic plasticity. The learning rules are implemented in a neural network, and evaluated on artificial data and simple real data (4 digits from MNIST dataset).

Qualitative Assessment

Overall, this is a decent paper that starts from first principles, then goes all the way to derive learning rules, which are then implemented in a neural network and tested on artificial and real data. This "completeness" is a strength of this paper. The novelty lies in not requiring normalization to account for gain variation, which is interesting. However, the weakness is in the evaluation. The datasets used are very simple (whether artificial or real). Furthermore, there is no particularly convincing direct demonstration on real data (e.g. MNIST digits) that the network is actually robust to gain variation. Figure 3 shows that performance is worse without IP, but this is not quite the same thing. In addition, while GSM is discussed and stated as "mathematically distinct" (l.232), etc., it is not clear why GSM cannot be used on the same data and results compared to the PPG model's results. Minor comments (no need for authors to respond): - The link between IP and the terms/equations could be explained more explicitly and prominently - Pls include labels for subfigures in Figs 3 and 4, and not just state in the captions. - Have some of the subfigures in Figs 1 and 2 been swapped by mistake?

Confidence in this Review

2-Confident (read it all; understood it all reasonably well)